# The Perceived Impact of Suicide Bereavement on Specific Interpersonal Relationships: A Qualitative Study of Survey Data

**DOI:** 10.3390/ijerph16101801

**Published:** 2019-05-21

**Authors:** Valeriya Azorina, Nicola Morant, Hedvig Nesse, Fiona Stevenson, David Osborn, Michael King, Alexandra Pitman

**Affiliations:** 1UCL Division of Psychiatry, Maple House, 149 Tottenham Court Road, London W1T 7NF, UK; valeriya.azorina.14@ucl.ac.uk (V.A.); n.morant@ucl.ac.uk (N.M.); hedvig.nesse.14@ucl.ac.uk (H.N.); d.osborn@ucl.ac.uk (D.O.); michael.king@ucl.ac.uk (M.K.); 2UCL Research Department of Primary Care & Population Health, Rowland Hill St, London NW3 2PF, UK; f.stevenson@ucl.ac.uk; 3Camden and Islington NHS Foundation Trust, St Pancras Hospital, London NW1 0PE, UK

**Keywords:** suicide bereavement, qualitative research, grief, bereavement, social support, interpersonal relationships, kinship

## Abstract

People bereaved by suicide have an increased risk of suicide and suicide attempt, yet report receiving less support than people bereaved by other sudden deaths. Reductions in support may contribute to suicide risk, yet their nature is unclear. We explored the impact of suicide bereavement on the interpersonal relationships of young adults in the UK using an online survey to collect qualitative data. We conducted thematic analysis of free-text responses from 499 adults to questions capturing the impact of bereavement on relationships with partners, close friends, close family, extended family, and other contacts. We identified four main themes describing the changes in relationships following the suicide: (1) Social discomfort over the death (stigma and taboo; painfulness for self or others to discuss; socially prescribed grief reactions); (2) social withdrawal (loss of social confidence; withdrawal as a coping mechanism); (3) shared bereavement experience creating closeness and avoidance; (4) attachments influenced by fear of further losses (overprotectiveness towards others; avoiding attachments as protective). These findings contribute to understanding deficits in support and pathways to suicidality after suicide bereavement. Such disrupted attachments add to the burden of grief and could be addressed by public education on how to support those bereaved by suicide.

## 1. Introduction

The World Health Organization (WHO) has committed to reducing the global suicide rate by 10% by 2020 in a bid to prevent a proportion of the estimated 800,000 deaths by suicide annually [1]. Although previously the WHO had estimated that each suicide left behind approximately 10 bereaved friends and family members [2], empirically-derived estimates suggest figures of between 60 [3] and 135 [4]. People bereaved by suicide are at risk of a range of adverse mental and physical health outcomes, including an increased risk of suicide [5,6], suicide attempt [7], depression [5], and psychiatric admission [8]. Their risk of suicide [5,6] and of suicide attempt [7] applies to non-genetically related contacts (spouses and friends), as well as blood relatives, indicating that the psychological (environmental) effects are likely to have wide reach within the social networks of the deceased.

Social support is usually regarded as having a buffering effect against adverse mental health outcomes following negative life events [9]. However, people bereaved by suicide are more likely to describe receiving less support from family and friends than people bereaved by any other causes of sudden death, as well as delays in receiving support [10]. They are also more likely to report poor social functioning than people bereaved by other unnatural deaths [7]. The impacts extend into educational and occupational functioning, with people bereaved by suicide being 80% more likely than people bereaved by sudden natural causes of death to drop out of a job or a course [11]. The reasons for these associations are unclear but quantitative studies comparing scores on dimensions of grief find that people bereaved by suicide report higher scores on rejection, shame [8], and perceived stigma compared to those bereaved by other unnatural deaths [12]. Qualitative research also suggests that the social awkwardness of friends and colleagues contributes to difficulties with social and occupational functioning [11,13,14].

To obtain an in-depth understanding of the nature of perceived changes or inadequacies in informal support following suicide bereavement, and the specific relationships it affects, we analysed qualitative data elicited from open questions in an online survey. Our objective was to explore the views of young adults bereaved by suicide on any changes in their relationships with family and friends since the bereavement, and the nature and quality of informal support available after suicide bereavement. In doing this we also hoped to understand the ways in which changes in social support from specific members of the social network might contribute towards mental health outcomes following suicide bereavement.

## 2. Materials and Methods 

### 2.1. Study Design and Participants 

We used a national cross-sectional study design and a closed online survey to collect qualitative data from a nested sample of people bereaved by suicide, drawn from a wider sample of people bereaved by sudden death—the University College London (UCL) Bereavement Study. We focused on young adults (aged 18–40), as this was the suicide risk group of greatest concern at the time of designing the study [15], with an observed tendency not to seek help for mental distress [16]. We chose not to recruit from bereavement support groups in order to avoid the biases associated with recruiting a help-seeking sample. We used the global email systems of 37/164 (20%) United Kingdom (UK) higher education institutions (HEIs) to recruit young adults aged 18–40 who were working or studying there, and had experienced the sudden bereavement of a close friend or relative since the age of 10. Although the WHO use an upper age limit of 24 for young adults [17], we chose to extend this to avoid collecting only recent experiences of support. Our sampling strategy has been described previously [7], and HEIs included a varied range of arts colleges, universities, and vocational training institutions (see Acknowledgements) sampled in 2010. This provided a sample of 659,572 staff and students. There was no accurate way of measuring response rate, as the denominator of bereaved people was not ascertainable using routine data or survey methods. All participants were invited to take part in a survey of “the impact of sudden bereavement on young adults”, with the survey link provided in the sampling email. All participants provided online informed consent, and were provided with a list of support sources.

The study protocol was approved by the UCL Research Ethics Committee in 2010 (reference: 1975/002).

### 2.2. Procedures

An online questionnaire was designed, refined, and piloted as described previously [18]. Questionnaire design was guided by the input of a consultation group of bereaved adults and bereavement counsellors, who advised on key domains to cover. Following the quantitative component of the questionnaire (119 fixed-response questions), 20 open questions elicited free-text data [7]. These probed specific domains likely to be impacted by the bereavement reported, and were worded to be neutral and non-leading, to avoid assuming solely negative outcomes of bereavement. Five questions inquired about the impact of the bereavement on a range of interpersonal relationships:


*“In what way, if any, has your relationship with a partner, or with potential partners, changed since the bereavement?”*



*“What about relationships with close friends, or with potential close friends?”*



*“In what way, if any, have relationships within your immediate family (parents, brothers, sisters, children) changed since the bereavement?”*



*“What about relationships with members of the wider family (cousins, aunts, uncles, nephews, nieces, grandparents)?”*



*“If there are other ways in which you have withdrawn from those around you or grown closer to them, please use this space to give details”.*


Responses had no upper word limit, with instructions to provide as much or little detail as wished, or to skip any question that did not apply.

### 2.3. Data Analysis

We analysed our primary data using principles of thematic analysis [19], facilitated using NVivo software [20]. We conducted our analytic work collaboratively as a group (researchers Valeriya Azorina, Hedvig Nesse, Alexandra Pitman, and Nicola Morant), holding regular reviews and discussion meetings to encourage reflexivity and enhance validity by providing opportunities to question and refine our interpretations and analytic processes [21]. Given the nature of our survey data (large numbers of respondents and low volumes of data per respondent), our analysis includes some exploration of how responses and inductively-derived themes were patterned across the five different questions in quantitative terms, as well as in-depth explorations of meanings and content that are more typical of thematic analysis. We adopted a staged approach. Initially, respondents were categorised (by Valeriya Azorina) into two broad descriptive categories, depending on whether or not they reported changes in any one of the five relationship categories probed (partners and potential partners; friends; immediate family; wider family; general relationships). This categorization was regardless of the level of detail provided.

Having agreed on this categorisation, we then proceeded to more in-depth data-driven analysis, developing codes and themes to capture aspects of content and how respondents described their experiences across all the questions. For this stage of the analysis, two researchers (Valeriya Azorina, and Hedvig Nesse) initially coded a sample of data from 100 participants independently, in order to develop a data-driven coding framework. In this second stage brief responses (such as “no change”, or “closer to dad”) were not analysed, as beyond these basic statements, they provided few insights into participants’ reasons for, or impact of, changes. As data analysis progressed, we conducted further cycles of coding and reflective team discussions that included reviews of the consistency and face validity of codes. Following Braun and Clarke’s approach to thematic analysis [19], this led to the emergence of higher-order themes and sub-themes, through the organisation of descriptive “close to the data” codes. A collaborative team-based analytic process was designed to enhance validity of findings by ensuring high levels of reflexivity and fostering conceptual thinking throughout the process [21]. Our team discussions ensured that we felt confident of the conceptual stability of the final themes and sub-themes we report below. We followed COREQ (Consolidated criteria for Reporting Qualitative research) guidelines [22] in presenting findings. Where quotes from online typewritten responses were included, we corrected spelling only where characters were omitted or inverted.

## 3. Results

Of the estimated 659,572 bereaved and non-bereaved people receiving the email invitation, 5085 people responded to the questionnaire by clicking on the survey link, and 4630 (91%) consented to participate in the online study. Of the 3432 participants meeting inclusion criteria, a total of 614 reported bereavement by suicide, of whom 499 (81%) responded to any of the five free-text questions on interpersonal relationships, and were included in the current analysis.

### 3.1. Participant Characteristics

Our sample was predominantly female (n = 416/499; 83%) and of white ethnicity (n = 461/499; 92%). The mean age of participants was 25 (standard deviation (SD) = 5.9), reporting bereavement at a mean age of 20 (SD = 5.6). Time since bereavement ranged from two weeks to 26 years (median = 4 years; IQR = 1–7). The majority (n = 354/499; 71%) had been bereaved by the death of a male. Over half (53%) reported being a blood relative of the deceased, and the remainder reported being a friend (33%), partner (4%), or other connection (9%), including in-laws, step-parents, colleagues and ex-partners.

### 3.2. Frequencies of Reported Changes across Relationship Types

The proportions of participants providing responses describing changes in specific relationships, in descending frequency, were relationships with close family (n = 427; 86%), friends (n = 414; 83%), wider family (n = 381; 76%), partners (n = 322; 65%), and relationships more generally (n = 264; 53%). A small minority (12/499; 2%) of all participants reported no changes in any of their relationships since the bereavement. When explored by relationship type, there was no clear pattern in those described as unchanged, which were most frequently relationships with wider family members (n = 134/381; 35%), close family (n = 71/427; 17%), friends (n = 63/414; 15%), and partners (n = 35/322; 11%).

### 3.3. Qualitative Themes

Our more in-depth analysis of responses describing the nature of, explanations for, and consequences of relationship changes after suicide bereavement produced four main themes, with associated sub-themes. For the most part these were conceptually distinct, and we highlight below where there are overlaps with, or differentiations from, other related or similar themes. The theme of social discomfort over the death applied to data from 118/499 of respondents (24%); the theme of social withdrawal to 212/499 (43%); that of shared bereavement experience creating both closeness and avoidance to 105/499 (21%); and that of attachments influenced by fear of further losses to 180/499 (36%). To provide an indication of how themes were distributed across specific relationships in this large sample, we report the number of respondents with data coded under each main theme in relation to each relationship type. All four themes applied to data describing each of the five relationship types, except for social withdrawal, which did not apply to wider family members but was the most commonly occurring theme in the sample overall (Appendix A).

#### 3.3.1. Social Discomfort over the Death 

Respondents often described the social discomfort of their relatives, friends and acquaintances on the topic of the suicide. This was described mostly for relationships with friends and close family (with 35% and 30% of all responses coded under this theme in relation to those relationship categories), followed by wider family (17%), general relationships (10%) and partner or ex-partner (8%). It was commonly recounted that relationships had become strained, and this was perceived to be due to others’ general discomfort with the notion of suicide. The awkwardness of these encounters appeared to reduce respondents’ sense of belonging within their usual social networks. Whilst this theme was identified in responses from fewer people than that of social withdrawal (see Appendix A), it was a theme that applied across all relationship types, indicating that social discomfort over the topic of suicide was pervasive across a range of social interactions.

##### Stigma and Taboo

Social awkwardness around the subject was often described in relation to the stigma and taboo around suicide, which for many felt very shameful. This often led people to avoid the topic or conceal the cause of death.


*“We (my immediate family) just never talked about it. It’s like the elephant in the room.”*
—Twenty-three-year-old female whose uncle had died by suicide 6 years prior.


*“The most overwhelming feeling was not guilt, but shame and embarrassment that this had happened, I did not want the awkward reactions and shocked reactions of people [close friends] that do not know what to say when you mention suicide.”*
—Twenty-year-old female whose father had died by suicide 2 years prior.

Many had learned to avoid the topic of the suicide because of previous negative experiences of others’ stigmatising attitudes or a lack of support and understanding.


*“I only told my best friend, who did not know how to react, and I felt awkward. This uncomfortable feeling I did not want to repeat so never spoke closely to friends about the suicide.”*
—Twenty-year-old female whose father had died by suicide 2 years prior.

##### Painfulness for Self or Others to Discuss

Discomfort in talking about the suicide also arose because it felt too painful and upsetting to discuss, either for the person themselves, or for others.


*“At first I did not tell any of my close friends or family because it was still too painful to talk about. Some of them still do not know. I am still telling people as I meet up with them. I have only recently (8 months later) been able to talk about it without crying.”*
—Twenty-year-old female whose close friend had died by suicide 8 months prior.

Where others also grieved the same loss, avoidance of the topic was sometimes viewed as a necessary means of preventing them from becoming distressed. Knowing whether or not to bring up the issue could be difficult for some.


*“I cannot discuss (it) with (my immediate family) as it upsets them, and I do not want to make them upset.”*
—Twenty-two-year-old female whose classmate had died by suicide 9 years prior.


*“It can be difficult with (close friends) who knew him too. Sometimes somebody wants to talk about it, sometimes not. It is hard to gauge.”*
—Thirty-year-old female whose close colleague had died by suicide 3 years prior.

##### Socially Prescribed Grief Reactions 

Respondents also described social awkwardness around expectations for how grief was managed and expressed. They sometimes felt others judged or felt uncomfortable about how they expressed their grief. This worked in two directions: either social pressure to “get over it” quickly and embarrassment over outpourings of emotion; or disapproval over not displaying “enough” emotion. Many respondents appeared highly attuned to these dynamics and found achieving this balance difficult because the rules governing socially prescribed grief behaviours were not always clear. These difficulties were described in several close existing relationships with family and close friends, and for some, were described as having a potentially lasting impact on these relationships. Whilst some felt reproached for wanting to talk about the deceased, others felt burdened by an expectation to share their feelings more openly.


*“After my partner passed I was treated different, not wanting me to talk about it, all traces gone from families’ houses and our house cleared out. But I think that was what my family thought I wanted, but I didn’t. I realise now I could not and will not be able to talk to family about my issues again.”*
—Thirty-three-year-old female whose partner had died by suicide 9 years prior.


*“I think that my friends expected me to cry and wail as it was all pretty horrific what had happened, but I just did not feel like that, I did a fair share of crying on my own, really did not feel the need to share that, there is this culture of talking and talking, but I did not feel there was anything to talk about, it was terrible and awful, yes, but then what?”*
—Thirty-eight-year-old female whose mother had died by suicide 10 years prior.

#### 3.3.2. Social Withdrawal 

Many participants reported that they had become more withdrawn from social relationships as a result of the bereavement. For some this was an adaptive coping mechanism, whilst for others this was described as a negative change relating to a loss of self-esteem, disappointment in others’ reactions, or a sense of being a burden to others. Most responses coded under this theme described a withdrawal from close friendships (39%), existing or potential romantic relationships (23%), primarily due to lack of trust and emotional withdrawal, or a general withdrawal from people around them (28%). Withdrawal from close family was less frequently reported—10% of responses in this category were coded under this theme. Such withdrawal was distinct from that influenced by fear of further losses, as described in theme 4.

##### Loss of Social Confidence

The majority of codes under this theme described social withdrawal as a negative outcome of their bereavement, relating this to having become more insecure, less outgoing, lacking self-esteem, and having difficulty making friends. Many described avoiding social events, especially in large groups of people, even if they had previously enjoyed these types of social situations or considered themselves to be quite “sociable”. A difficult combination of feelings was often described as underlying this process. Despite withdrawing, several people mentioned a simultaneous longing for connection.


*“I tend to withdraw myself from others (people in general) most of the time whilst at the same time feeling a need to be really close to them.”*
—Eighteen-year-old female whose close friend had died by suicide 5 years prior.

Explanations for a loss of social confidence primarily revolved around disappointment in others over their perceived lack of support or understanding of grief or mental health difficulties. Some respondents described feeling let down by friends’ reactions, with knock-on effects on their trust in others. Perceptions of being a burden to others because of their grief, and of feeling disliked or rejected, were also described as common drivers of self-isolation.


*“I had a couple of friends who, after a few months didn’t have space for my grief or PTSD, which meant I trusted them less and closed off a bit from them.”*
—Twenty-eight-year-old female whose close friend had died by suicide 5 years prior.


*“I often feel like I am a burden to my friends if I vent my grief.”*
—Twenty-two-year-old female whose close friend had died by suicide 6 years prior.

There was a sense of a reciprocal relationship between these two sub-themes, such that self-isolation as a coping mechanism may have reinforced others’ distance, and that perceptions of lack of support from others may have reinforced self-isolation, as well as reducing self-esteem and social confidence.


*“I feel the need to be close to my friends, but they just seem to push me away.”*
—Eighteen-year-old female whose close friend had died by suicide 5 years prior.

##### Withdrawal as a Coping Mechanism

A minority of codes under this theme described using withdrawal as a means of coping with grief by making deliberate decisions to withdraw from relationships and spend more time alone. They reported needing more time or space for themselves, often as a way of processing emotions privately.


*“I find that occasionally I get very low and have to retreat away from (people in general) in order to vent anger, hurt, frustrations.”*
—Twenty-six-year-old female whose mother had died by suicide 7 years prior.


*“Definitely withdrew at times (from relationships more generally). There was no way that I found to express my grief other than at times taking myself away thinking about him and crying. There was no way to get the feelings out other than that, no rationalisation because there was nothing to rationalise.”*
—Twenty-one-year-old male whose close friend had died by suicide 3 years prior.

#### 3.3.3. Shared Bereavement Experience Creating both Closeness and Avoidance

Shared experiences of bereavement were frequently described as a means of identifying who could show empathy, often bringing the bereaved closer to people with similar experiences. However, having a shared experience of bereavement was also seen as risking being exposed to another’s intense grief, and this was cited as a reason for distancing oneself from others who were grieving the same loss. Half of responses coded under this theme referred to relationships with friends (51%), and 25% referred to relationships with close family, followed by general relationships (16%). A shared bereavement experience was rarely mentioned as a factor influencing the closeness of relationships with wider family (5%) or romantic partners (3%).

##### Closeness to Those with Shared Experiences of bereavement 

For many, a shared bereavement experience had become an important influence on their affinity to others. This had either positive or negative effects on specific relationships; closeness to those with similar experiences, and a distancing from those with no understanding of the situation. Reasons cited for becoming closer to people who had experienced bereavement, particularly suicide bereavement, were that they did not require explanations, could relate to their pain, and knew how to comfort them appropriately. Reasons for becoming less connected to those with no experience of bereavement were that they could not understand the depth and complexity of emotions consequent to a suicide loss. This sub-theme was therefore distinct from that of a more general social withdrawal (whether due to loss of confidence or as a coping mechanism) because social connection relied on having a shared bereavement experience.


*“Much closer to friends who have also lost a parent and closer to an older friend in particular whose daughter died at 15 from a brain tumour. She understood things like picking out coffins, anniversaries, etc., which friends my own age did not have experience of.”*
—Twenty-two-year-old female whose mother had died by suicide 1 year prior.


*“I find I am drawn to (people) who have lost someone to suicide. It is like I can pick them out in a crowd. I find I can talk to them about everything in my life, and they are instant friends and confidants.”*
—Twenty-eight-year-old female whose partner had died by suicide 5 years prior.

Whether or not friends and relatives had known the deceased was also reported as a factor influencing the closeness of those relationships after the loss, partly due to the shared bereavement experience. Where the deceased was known by others in the social network, affinity to others was also influenced by whether those individuals had a similar coping strategy. Respondents described having become closer to those who handled their grief similarly, but less close to those who managed it differently.


*“Another friend and I coped very similarly straight after the death of our friend, and so became inseparable in the weeks and months afterwards and remained close friends for years afterwards.”*
—Twenty-year-old female whose close friend had died by suicide 7 years prior.


*“We (my immediate family) all seemed to deal with the death differently and probably ran in different directions; this may be the reason why we are not close, we did not deal with it together as a family.”*
—Thirty-four-year-old female whose mother had died by suicide 22 years prior.

##### Avoidance of Those with Shared Experiences 

Our analysis also identified cases in which bereaved individuals found the shared experience of bereavement too intense, regardless of any similarities in coping styles. In contrast to those who felt affinity to other bereaved people, these individuals explained that they preferred the company of people who had not known the deceased. This was because they found it hard to spend time with friends or family grieving the same loss, whether close or more peripheral contacts, perhaps because the expectation of reciprocal support felt too burdensome.


*“I became a lot closer to certain friends, predominantly those who were not directly involved in the bereavement, as I found it easier to talk to them as they were not having to deal with all the emotions too. Those who were directly involved, I found it harder to speak to them and tried to avoid them.”*
—Nineteen-year-old female whose ex-partner had died by suicide 3 years prior.

This creation of social distance appeared to be motivated by self-protection against others’ visceral grief, particularly people less well known to them. In the section that follows we discuss how social distances were established in existing and new relationships, but with the different motivation of protecting against future abandonments.

#### 3.3.4. Attachments Influenced by Fear of Further Losses

Respondents described having developed a range of fears about the potential for future losses, which seemed to have affected their attachment behaviour and the manner in which they conducted their relationships. Such fears primarily related to close relatives dying, with a specific fear of further suicides, but also a general dread of abandonment within friendships and romantic relationships. Most commonly these fears related to close family (40% of responses coded under this theme) or romantic partners (31%), and less often to friends (12%), people in general (12%) or wider family (5%).


*“I am much more scared of those I love (partners or potential partners) dying or abandoning me.”*
—Twenty-four-year-old female whose sister had died by suicide 3 years prior.


*“Now I am more aware of suicide, I worry each time I get into an argument (with my partner). I am scared of suddenly losing him.”*
—Twenty-five-year-old female whose brother had died by suicide 8 years prior.

This dread influenced behaviour within relationships in two broad ways. Some described a conscious awareness of having become more attentive and protective of others (primarily close family and partners), whilst others reported efforts to maintain a distance in order to protect themselves against abandonment (primarily by close friends).

##### Overprotectiveness towards Others

Those who became more protective of others described the strain of keeping partners or relatives happy or avoiding upsetting them, as part of preventive efforts. By maintaining vigilance for symptoms of depression in close relatives, and lowering their threshold for offering help, they hoped to be able to intervene as early as possible. The greater intensity of these relationships appeared to be associated with quite significant emotional strain, on top of the work of grief. Any additional closeness gained from these relationships with partners or close relatives was often therefore at some cost.


*“Anxious about making him (partner) happy and making sure he would not get depressed like my uncle.”*
—Twenty-nine-year-old female whose uncle had died by suicide 3 months previously


*“I am more quick to offer help and comfort when they (partner or potential partners) have depressing feelings.”*
—Twenty-year-old female whose close friend had died by suicide 5 years prior.

There was a sense in which this sub-theme applied primarily to family and partners because of a greater sense of responsiblity for their well-being, and because the prospect of their loss would be the most devastating. However, where this sub-theme applied to friendships, there was evidence that respondents were proactive in helping out friends with mental illness, monitoring friends’ well-being, and offering help even to acquaintances.

##### Avoidance of Attachments as Protective 

Those who reported efforts to maintain a self-protective distance from others commonly described this with respect to new friendships or romantic relationships, as part of an effort to avoid further experiences of rejection or potential loss. This appeared to be particularly marked in relation to people perceived as vulnerable or at risk of suicide.


*“I am extremely conscious of being independent and not getting too close to people (friends or potential close friends) for the fear that I will lose them.”*
—Twenty-three-year-old female whose mother had died by suicide 12 years prior.


*“I find it difficult to maintain a friendship or make a friendship with people who have depression or a related mental health problem out of fear that the same thing might happen again.”*
—Twenty-year-old female whose close friend had died by suicide 4 years prior.

This sub-theme applied more specifically to people whom respondents were currently less close to, in contrast to the previous sub-theme of overprotectiveness towards others, which applied to both future and current close relationships with partners and close relatives. It was also distinct from the sub-theme of using social withdrawal from existing relationships as an emotional coping mechanism, in that fear of hypothetical future events was central. However, in both these themes there were similar consequences in reinforcing or perpetuating social isolation.

## 4. Discussion

### 4.1. Main Findings 

Our thematic analysis of 499 participants’ online responses found that it was common for people bereaved by suicide to perceive changes in their interpersonal relationships after the loss. These changes were predominantly experienced as negative, with all five categories of relationships becoming more strained or distant, whether due to the stigma of suicide, the social awkwardness of others in discussing the death, or others’ lack of support and understanding. The majority of changes perceived were in family relationships and close friendships. For family relationships the most prominent theme was social withdrawal, whilst for close friendships, it was fear of further losses. For those who described changed relationships with partners, both social withdrawal and fear of further losses were prominent. In relation to wider family, social discomfort was the most mentioned theme but social withdrawal did not feature at all, possibly because these relationships were already less intimate than others. The key distinguisher of whether a relationship would become closer or more distant was initial closeness. Thus, family relationships were often affected by overprotectiveness due to a fear of other losses, while friendships were often characterised by social withdrawal and discomfort.

Social withdrawal from others, as the most common theme across all relationship types, was largely driven by a loss of social confidence or processing of grief, but was sometimes accompanied by a longing for connection. In some cases, there seemed to be a reciprocal relationship between self-isolation and others’ withdrawal, which reinforced stigma and shame, and reduced social confidence. There was little sense of control over changes in relationships, which left many feeling isolated and rejected. However, some had made a conscious decision to distance themselves from others as an emotional coping strategy or a form of self-protection from further losses or from others’ grief. Therefore, a shared experience of suicide bereavement did not always offer valued peer support, due to wariness about being swamped by others’ grief.

Positive changes observed included a strengthening of emotional bonds with other bereaved people who could comfort them appropriately or who had similar coping strategies. Increased closeness sometimes came at an emotional cost, where it arose out of overprotectiveness or hypervigilance for signs of depression or suicidality in others. These different grief coping styles, therefore, led to characteristic changes in specific relationship types, demonstrating the importance of investigating this question at this fine-grained level.

### 4.2. Findings in the Context of Other Studies

Comparing these findings to other British work, two qualitative interview studies have investigated post-suicide support experiences more widely, including the informal support received by family and friends [11,23]. In both those studies, as in the current one, themes of stigma and the pressure to conceal one’s grief, to express it in “appropriate” ways and within a limited time period were identified, and appeared to place a strain on social relationships. As those studies also included people bereaved by non-suicide violent deaths, such pressure to conform to socially prescribed grief behaviour appears to be experienced by both groups [11,23]. Such group similarities are also apparent internationally. In a Norwegian qualitative study of people bereaved by suicide, accidental death, and sudden infant death syndrome, participants reported that because these were unusual deaths, others were unsure about what to say or how to offer support [24]. This is contrasted with a Norwegian qualitative study describing the experiences of people bereaved by terrorist attack, in which public outrage appeared to fuel an outpouring of support [25]. Whilst this did not preclude experiences of others’ avoidance and social awkwardness, it was comforting for participants to know that other offers of support were available.

Our own previous analysis of interviews with people bereaved by suicide and other sudden deaths (as here, a nested sample from the same wider British sample) also identified the social awkwardness of others as a key dimension of perceived stigma of sudden bereavement, creating difficulties in social relationships [11]. As in the current study, people bereaved by suicide described their friends and relatives avoiding the topic of the suicide, and avoiding them socially, leaving them feeling unsupported. Such findings are similar to those of qualitative work carried out in Australia, showing that people bereaved by suicide described experiences of stigma and insensitive comments about suicide as a barrier to support [26,27], and perceived shame and self-blame in relation to social withdrawal and loss of social support [26]. An Australian study of people bereaved by suicide also identified a perception of taboo and stigma around suicide within the family [28]. This resonates with the sub-theme we identified of social discomfort relating to the stigma and taboo around suicide. This is also consistent with a number of literature reviews finding stigma to be a particular issue for people bereaved by suicide [8,29,30,31,32]. Such work identifies stigma as associated with social ostracism, and self-stigma as associated with self-isolation and withdrawal [29,32].

Our sample was relatively young, and our findings regarding the affinity of people with shared bereavement experiences are similar to those of two Australian qualitative studies with bereaved adolescents [33,34]. These found that young people were drawn to others with similar bereavement experiences [33], preferring to disclose their bereavement to a limited number of contacts, and placing great value on feeling accepted [34].

Our theme of social withdrawal supports theoretical work arguing that whilst it is common for bereaved people to disengage socially and isolate themselves, some people may perceive their implicit needs for attachment as unmet, which in turn may leave those individuals feeling deserted and helpless [35]. Such reinforcement resonates with our observation of a feedback loop for some individuals between self-isolation and others’ withdrawal. However, for other bereaved people a period of social withdrawal may form part of their grief work, and may only be perceived as problematic if it becomes persistent or difficult to reverse. Qualitative work from Switzerland notes that it is the individual coping style of an individual bereaved by suicide that influences the extent to which others’ help is mobilized [36]. This was not apparent in our dataset, but would be worth exploring further using interview methods, and in people bereaved by different causes of death.

### 4.3. Strengths and Limitations

To our knowledge, this is the only qualitative study of the specific impact of suicide bereavement on different interpersonal relationships. By collecting data from a large defined national sample of HEIs, we avoided using a help-seeking sample, and instead identified a more representative sample of young bereaved adults than other studies of support experiences. All participants were presented with the same five neutrally-worded open questions, creating a rich dataset probing the specific impact of bereavement on various interpersonal relationships. This, and the large sample size, enabled us to report on the relative frequencies of our qualitative themes and how these were distributed across data relating to different types of relationship. The specificity of the questions meant that we were able to explore how relationship dynamics varied between different types of relationships, including established and new. However, caution should be exercised in interpreting frequency data, which is best considered as indicative of variations in, and dominance of, themes. Regular discussions between a team of four researchers representing clinical and non-clinical backgrounds throughout the analytic and reporting process enhanced reflexivity and rigour [37].

Our analysis identified clear commonalities in themes, with this and our sampling approach, increasing the likelihood that our findings may be relevant to others bereaved by suicide. Our use of an internet-mediated approach to sampling and data collection may have reduced social desirability effects and improved disclosure through its anonymity. However, sampling through UK HEIs may have introduced selection bias in relation to higher socio-economic groups and the healthy worker effect [38]. We acknowledge the predominance of white females in our sample, most of whom had lost a male friend or relative (most commonly father, uncle, cousin, or brother) to suicide, in keeping with the age structure of suicide risk. Our sampling of young adults meant that we could not capture the impact of suicide loss on parents’ or grandparents’ relationships. The impact of suicide bereavement on interpersonal relationships of men and specific ethnic groups warrants further exploration. We also acknowledge that our dataset included the experiences of those only recently bereaved and those bereaved for many years, and that whilst this provides insights into the longitudinal trajectory of relationship changes, the majority had been bereaved for over a year. This both introduces the potential for recall bias, and under-represents the experiences of the recently bereaved.

### 4.4. Clinical and Policy Implications

The disrupted attachments described by some in this study are concerning in a group at elevated risk of suicide [5,6] because of the emotional burden they add to the grief process. Our findings illustrate how diverse changes in interpersonal relationships might influence mental health. The themes identified contribute to our understanding of the associations of suicide bereavement with poor social and occupational functioning [7], greater perceived stigma [12], and a lack of informal support [10]. Some individuals in our study, as in other studies [39,40], reported deriving great value from the support of other bereaved people. Others found it hard to be with other bereaved people unless their coping styles were similar. Quantitative evidence suggests that mutual support (bereavement support groups, individual peer support) is associated with prolonged psychological difficulties, perhaps due to ruminative coping styles [41]. Our study therefore highlights the range of support that should be available to this group. The stigma and awkwardness described after suicide bereavement is likely to limit the availability of informal support but also the bereaved person’s willingness to seek or accept support. This may explain perceptions of a lack of informal support [10].

Given these deficits in informal support, there is a role for general practitioners, bereavement counsellors, managers, and teaching staff to be more proactive in offering the option of support to someone bereaved by suicide to ensure equitable access. Consultation with suicide-bereaved people reveals a clear demand for early access to formal support, offered proactively to overcome the barriers to help-seeking created by self-stigma [39,40]. However, such offers should recognize that many of the processes described in this study (social withdrawal and other relationships changes) may be part of an adaptive mourning process after traumatic loss; a process that may take several years. A number of countries have already developed systems of proactive early intervention after suicide [42]. However, further work is needed to establish how to tailor services to different age, gender, and ethnic groups. Such services will need to avoid pathologising grief, and also respect the wishes of some bereaved people to withdraw from others and cope by themselves [10]. Online forums tailored to those bereaved by suicide have the advantage of anonymity, combining psycho-educational resources with a sense of belonging, and may be an entry point to further face-to-face support. Campaigns to improve public awareness of support services for people bereaved by suicide will increase the chances that people who experience suicide bereavement will know how to access support, and that those who have contact with them can recommend services. Campaigns could also tackle the taboo and stigma around suicide, and educate the public about how this stigma can distress the bereaved and how to provide appropriate informal support after a suicide. This, in turn, would reduce self-stigma in the bereaved, and the barriers this creates to accessing formal and informal support.

### 4.5. Future Research 

Although this study provides a deeper understanding of interpersonal dynamics after the experience of suicide bereavement in one British sample, further studies are needed to explore the experiences of men and women in other age groups and cultures to inform the design of tailored, culturally-sensitive interventions. Given the prominent theme of social withdrawal it would be valuable to evaluate the role of online community forums, and their effect on mental health, grief and social outcomes. Qualitative work to assess the acceptability of these interventions and others would ascertain the demand for online [43], face-to-face, group, and individual interventions for people bereaved by suicide.

## 5. Conclusions

Relationships with friends, partners and relatives after suicide bereavement are often characterised by social awkwardness, others’ avoidance, perceived stigma, a lack of understanding, and a fear of further abandonments. Relationships with close friends and close family are those most often negatively affected. Special attention should be paid to a bereaved person’s experiences of changes in friendship dynamics, as social withdrawal was commonly reported in relation to friendships and disrupted attachments appear to add to the burden of grief for some. It is important to educate the lay public in how best to support someone in their network who is bereaved by suicide, and to raise awareness of the formal support options available.

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
