# Peer review of "The Perceived Impact of Suicide Bereavement on Specific Interpersonal Relationships: A Qualitative Study of Survey Data"

_ijerph, 2019, doi:10.3390/ijerph16101801_

Reviewer 1 Report

The manuscript presents a qualitative research in perceived changes in interpersonal relationships In those who lost someone by suicide. The study has methodological foundation and its findings are relevant.

Two minor details.

The first paragraph in the results section could be included in the materials and methods section.

State clearly if there were any criteria for excluding participants due to the quality of their answers.

Best wishes.

Author Response

IJERPH Manuscript ID  ijerph-490289

The perceived impact of suicide bereavement on specific interpersonal relationships: a qualitative study of survey data

Reviewer 1:

Yes

Can be improved

Must be improved

Not applicable

Does the introduction provide sufficient   background and include all relevant references?

(x)

( )

( )

( )

Is the research design appropriate?

(x)

( )

( )

( )

Are the methods adequately described?

( )

(x)

( )

( )

Are the results clearly presented?

(x)

( )

( )

( )

Are the conclusions supported by the results?

(x)

( )

( )

( )

The manuscript presents a qualitative research in perceived changes in interpersonal relationships In those who lost someone by suicide. The study has methodological foundation and its findings are relevant. Two minor details.

The first paragraph in the results section could be included in the materials and methods section.

This was not a re-analysis of data, but a primary collection of data to address a range of pre-specified research questions. The specific research question here was the impact of suicide bereavement on interpersonal relationships. We have clarified this in the methods, but we have retained the reporting of sample size and sample characteristics under Results to reinforce this point.

State clearly if there were any criteria for excluding participants due to the quality of their answers.

We omitted this in our submission – thank you for highlighting this and we agree it is important to explain our approach. We did not exclude responses on the basis of quality so much as interpretable content. That is to say that in our initial basic categorisation of responses by whether relationships had changed or not, brief responses (e.g. “no change”; “closer to dad”) were included. However, in the second stage of our analysis, which took a more fine-grained approach, we did not analyse data from these brief responses as they provided few insights into participants’ reasons for, or impact of, changes. We have added these points to the methods section to clarify this point.

Reviewer 2 Report

Thank you for providing me the opportunity to review this manuscript. The manuscript reports on a qualitative study looking at the perceived impact of suicide bereavement on the relationships of the bereaved individuals. The topic of the study is important as social support is considered to be important in coping with the loss. The relatively large sample (compared to other studies in this field) not drawn from a help-seeking population is a strength of the study. However, regrettably, despite its potential, the manuscript includes major flaws. I detail my concerns below.

1. While reading the results section, I was not convinced by the delineation of the four themes. I often thought that what I was reading, could equally have been placed with another (sub)theme. Also, the text includes several sentences highlighting that a specific theme was different from another theme, or conversely, that a theme overlapped with another theme. Sentences like these may indicate that the analysis of the material has not been finalized so that distinct themes could be created. Hence, I suggest that authors work further on the themes.

2. The survey concerned sudden death. This paper reports on suicide bereavement only. How come the other data were not included?

In the Discussion (p.10), authors refer to other studies which included other types of death, highlighting the importance of comparing outcomes of different types of death. Hence, limiting the current paper to suicide bereavement seems a missed opportunity to shed valuable light on social support after different types of death.

3. It is not clear when the study was conducted. It is also not clear if participants were instructed to think about a time frame when answering the questions listed on lines 99 to 107. There can be substantial differences in how people react to a bereaved individual immediately after the loss, or after 6 months, or one or two years, etc. Without knowledge of the time frame, or without time frame, the Results could be comparing apples with oranges.

4. I was surprised by the unbalanced description of the ‘Main findings’ in the Discussion. I had the impression that negative experiences were emphasized, with only few sentences describing positive experiences.

5. The section ‘Findings in the context of other studies’ could refer to other studies about sharing and social support in the context of grief.

6. In the Discussion (line 500) social withdrawal is linked to the survey, to mental illness, and then to suicide bereavement. Please rephrase to avoid stigmatization of suicide bereavement.

7. Similarly, lines 537 to 540 seem to exaggerate the study findings. The study findings do not speak about suicidality or mental illness in suicide bereavement. It is ironic that authors refer to ‘perceived stigma’ on the next line, as their interpretation of the findings may be seen as stigmatizing in itself.

8. On the next page (line 559) authors stated that services must avoid pathologizing grief. Still, in the same paragraph it is recommended that general practitioners and other professionals should screen for depression, anxiety, suicidal thoughts. They should differentiate between those who can cope on their own and others, which is obviously impossible. Unless I have misread it, this whole paragraph seems a strong plea to pathologize grief. The fact that grief (with or without social support) is a normal human reaction after a significant loss is completely ignored.

9. Overall, the whole discussion could be more balanced, looking at (potentially) negative and/or (potentially) positive experiences and their implications, in light of the broader literature.

10. Limitations include sample consisting predominantly of white females, and risk of recall bias (e.g. time since loss up to 26 years)

11. What is ‘stigmatising avoidance’?

Overall, I think that the study is important, and might entail important findings. Hence, I listed a few points that are amendable, and may contribute to the revision of the manuscript. Especially further work on the results and discussion could result is a much stronger manuscript. Good luck!

Author Response

IJERPH Manuscript ID  ijerph-490289

The perceived impact of suicide bereavement on specific interpersonal relationships: a qualitative study of survey data

Reviewer 2:

Yes

Can be improved

Must be improved

Not applicable

Does the introduction provide sufficient   background and include all relevant references?

(x)

( )

( )

( )

Is the research design appropriate?

( )

(x)

( )

( )

Are the methods adequately described?

( )

( )

(x)

( )

Are the results clearly presented?

( )

( )

(x)

( )

Are the conclusions supported by the results?

( )

( )

(x)

( )

Thank you for providing me the opportunity to review this manuscript. The manuscript reports on a qualitative study looking at the perceived impact of suicide bereavement on the relationships of the bereaved individuals. The topic of the study is important as social support is considered to be important in coping with the loss. The relatively large sample (compared to other studies in this field) not drawn from a help-seeking population is a strength of the study. However, regrettably, despite its potential, the manuscript includes major flaws. I detail my concerns below.

 1. While reading the results section, I was not convinced by the delineation of the four themes. I often thought that what I was reading, could equally have been placed with another (sub)theme. Also, the text includes several sentences highlighting that a specific theme was different from another theme, or conversely, that a theme overlapped with another theme. Sentences like these may indicate that the analysis of the material has not been finalized so that distinct themes could be created. Hence, I suggest that authors work further on the themes.

We have discussed this as a group and each author has gone back over the analysis to look for areas where delineation of themes could be refined. We had spent considerable time on the analysis and were confident that our themes and sub-themes were sufficiently developed and finalised as part of a thorough team-based analytic processes. This team approach had enhanced reflexivity through multiple team discussions about our emerging conceptualisations of themes, their working definitions and final names of themes and sub-themes. In response to these comments we have clarified the analytic process we adopted by adding some additional text to the Methods section describing Data Analysis (p3; lines 135-142), and to the qualitative analysis Results section on page 4 (lines 178-181).

We are aware, as R2 points out above, that there are some overlaps between themes, and we have described these as necessary in the text. Overlap of themes is an inevitable feature of rich qualitative data and its analysis using thematic (and other meaning-based forms of) analysis, as meanings co-exist in descriptions of complex subjective experiences and phenomena. Thematic analysis manages this co-existence, complexity and overlap through processes of multiple coding, in which a single piece of data may be coded in relation to several different analytic codes if it includes multiple related, or sometimes even contradictory ideas. The method recognises rather than seeks to simplify this complexity, whilst also enabling a final analytic product in which distinct themes can be described and presented in an accessible format. Thus, whilst for the most-part themes are presented in separate sections for clarity and ease of presentation, there are some points in the analytic text where we highlight comparisons with, or distinctions from other themes as appropriate. For example there were two points where we had specifically mentioned overlap: Section 3 of Results, re: ‘closeness to those with shared experiences of bereavement’ (p.8, lines 355-358), and at the end of theme 4 in the Results (p10, lines 463-469).

For the first of these (under the sub-theme - Closeness to those with shared experiences of bereavement; page 8, lines 355-358) we had mentioned: “This sub-theme therefore overlapped with the theme of social withdrawal due to difficulties opening up emotionally to people who had not experienced bereavement”. This may have been misleading, because the preceding sentence described how this sub-theme defined those who were drawn closer to other bereaved people as distinct from perceiving themselves as more distant from the non-bereaved. We have therefore clarified the text to indicate that the specific sub-theme of Closeness to those with shared experiences of bereavement related to social connections relying on shared experience, as distinct from more generic social withdrawal. Whilst this sub-theme captures the experience of feeling closer to others who have been bereaved, it also presents examples of the converse position of feeling somewhat distant from those without this experience. This therefore provides some internal validity to this sub-theme. 

For the second example, at the end of the Results section (p.10, lines 463-469), we have also added some text aiming to clarify the exact nature of these similarities and differences with other themes)

We trust that greater detail about our analytic processes, some additional text about the nature of relationships between themes in the text, and the information we have provided here reassure R2 that our analysis has been thorough and robust, whilst not making the error of assuming a “single truth”.

2. The survey concerned sudden death. This paper reports on suicide bereavement only. How come the other data were not included? In the Discussion (p.10), authors refer to other studies which included other types of death, highlighting the importance of comparing outcomes of different types of death. Hence, limiting the current paper to suicide bereavement seems a missed opportunity to shed valuable light on social support after different types of death..

Sampling for the survey described the project as investigating the impact of sudden bereavement. This masked respondents to the hypothesis, which was that outcomes (suicide attempt, social functioning, occupational functioning, stigma, shame, receipt of support) would be poorer in people bereaved by suicide than those bereaved by other causes of sudden death. The findings reported in our quantitative papers from this project supported those hypotheses, so our aim in this qualitative study was to understand why social support was perceived as lacking in this group. Understanding support needs in other groups is also interesting, but comparing their experiences would constitute a different research question and a different analysis, particularly given the very large dataset.  We therefore explored the specific issues this group face, rather than describing group differences. Our intention was that this would have relevance for implementation of suicide prevention strategies, which emphasise the support needs of the suicide-bereaved over other bereaved groups.

3. It is not clear when the study was conducted. It is also not clear if participants were instructed to think about a time frame when answering the questions listed on lines 99 to 107. There can be substantial differences in how people react to a bereaved individual immediately after the loss, or after 6 months, or one or two years, etc. Without knowledge of the time frame, or without time frame, the Results could be comparing apples with oranges.

We had omitted to say that the study was conducted in 2010, although mentioned that ethics approval was in 2010. We have now added the date of sampling to the methods. In the 5 questions about the impact of bereavement on relationships we did not ask participants to think about a time-frame. It may also have been very difficult for respondents to have established the timing of their various difficulties, which could have occurred at multiple points.  However, at the start of the questionnaire they were asked to relate their responses to all questions to their own experience of bereavement by suicide. If they had been bereaved by suicide more than once they were asked to answer the questions in relation to the loss of the person to whom they felt closest. Implicit to these instructions was the expectation that responses would cover experiences over the whole period since the loss. This ranged from weeks up to 26 years, and so for all quotes we qualified the time frame since the loss. We were keen to include a range of time frames to explore a range of experiences, and because we anticipated that some changes would take some time to become apparent. In our strengths and limitations we had indicated that this allowed us to look at the impact of suicide bereavement on relationships both established and new. However, we have also added to our limitations the following:

“We also acknowledge that our dataset included the experiences of those only recently bereaved and those bereaved for many years, and that whilst this provides insights into the longitudinal trajectory of relationship changes, the majority had been bereaved for over a year.”

4. I was surprised by the unbalanced description of the ‘Main findings’ in the Discussion. I had the impression that negative experiences were emphasized, with only few sentences describing positive experiences.

We agree that the results emphasise negative experiences and this is because quantitatively there were very few responses describing positive changes to relationships in our dataset. When analysed thematically, these positive experience were also not prominent. We were confident that we had not used leading questions in our survey, because these had been deliberately worded to be neutral, as per the input of our consultation group of those with lived experience. This avoided assuming solely negative outcomes of bereavement, and enabled both positive and negative responses. We mention this at the top of p3. The relatively few positive experiences reported would not therefore be explained by use of a question that had primed negative experiences. In summarising our findings we described what had been identified thematically in the data rather than presenting both sides with equal emphasis, as this would mislead readers by conveying a more positive overall picture than that observed in the original data. Another reason to explore the implications of negative impacts in detail in our Discussion was to suggest how to improve available support for the sub-group of bereaved people who receive inadequate informal support.

 5. The section ‘Findings in the context of other studies’ could refer to other studies about sharing and social support in the context of grief.

Thank you for this suggestion. This section was lacking, particularly in the international literature, and we have identified 8 key international studies that address this topic, most relating specifically to suicide, but others highlighting contrasts and similarities with groups bereaved by other cause. We have added the text to that section to widen the context for our findings, and feel it is much improved.  

6. In the Discussion (line 500) social withdrawal is linked to the survey, to mental illness, and then to suicide bereavement. Please rephrase to avoid stigmatization of suicide bereavement.

We have decided to cut down this paragraph as some parts went beyond what we can infer from the study findings. We also point out that social withdrawal may for some individuals be an adaptive stage in their grief work, and may not be perceived as problematic.

7. Similarly, lines 537 to 540 seem to exaggerate the study findings. The study findings do not speak about suicidality or mental illness in suicide bereavement. It is ironic that authors refer to ‘perceived stigma’ on the next line, as their interpretation of the findings may be seen as stigmatizing in itself.

As above we have cut down this paragraph as it went beyond what we can infer from this study, and acknowledged that our findings shed light on the observed quantitative associations with poor social functioning and a lack of informal support.

8. On the next page (line 559) authors stated that services must avoid pathologizing grief. Still, in the same paragraph it is recommended that general practitioners and other professionals should screen for depression, anxiety, suicidal thoughts. They should differentiate between those who can cope on their own and others, which is obviously impossible. Unless I have misread it, this whole paragraph seems a strong plea to pathologize grief. The fact that grief (with or without social support) is a normal human reaction after a significant loss is completely ignored.

We have edited down and limited our recommendations in this paragraph as per these comments, and balanced the discussion better to indicate that these processes may be normal for many people. Instead we have suggested that whilst professionals should be more proactive in offering support (to ensure that there is equitable access if needed) they should also recognise that many of the processes described (social withdrawal and other relationships changes) may be part of the natural mourning process after a traumatic loss, which is a process that may take several years.

9. Overall, the whole discussion could be more balanced, looking at (potentially) negative and/or (potentially) positive experiences and their implications, in light of the broader literature.

 These comments have helped us revise our Discussion to reflect the wider literature, the potential for recall bias, and to make the point that social withdrawal may be a normal part of the grief experience.  

10. Limitations include sample consisting predominantly of white females, and risk of recall bias (e.g. time since loss up to 26 years)

We had noted that this was a predominantly white female sample, most of whom had lost a male friend/relative to suicide, and this reflects the epidemiology of suicide. In this sample the most common form of suicide loss was of a close friend, father, uncle, cousin or brother ie young and middle-aged men. We have clarified this in the text under limitations. We agree that recall bias is a limitation of this study, and have added to this section that there is an issue regarding recall bias. Thus:

“We also acknowledge that our dataset included the experiences of those only recently bereaved and those bereaved for many years, and that whilst this provides insights into the longitudinal trajectory of relationship changes, the majority had been bereaved for over a year. This both introduces the potential for recall bias, and under-represents the experiences of the recently bereaved.”  

11. What is ‘stigmatising avoidance’?

This term was indeed unclear and we have amended this (in the Conclusion) to: others’ avoidance, and perceived stigma

Overall, I think that the study is important, and might entail important findings. Hence, I listed a few points that are amendable, and may contribute to the revision of the manuscript. Especially further work on the results and discussion could result is a much stronger manuscript. Good luck!

Round  2

Reviewer 2 Report

Thank you for considering my comments and the work authors have done for revising the manuscript. I was; however, surprised to read that authors think that: “Overlap of themes is an inevitable feature of rich qualitative data”. Overlap is not inevitable. Themes are created by authors, it is not something that emerges “inevitably”. If authors choose to retain their (sub-)themes, so be it. I have no further questions. Good luck.